

## 2 **Fractionation of stable carbon isotopes during formate consumption in**

## 3 **anoxic rice paddy soils and lake sediments**

Ralf Conrad[1], Peter Claus[1]
[1]Max Planck Institute for Terrestrial Microbiology, Karl-von-Frisch-Str. 10, 35043 Marburg,
Germany
Correspondence to: Ralf Conrad (Conrad@mpi-marburg.mpg.de)
**Running head:** Isotope fractionation by anaerobic formate consumption



**Abstract.**
Formate is energetically equivalent to hydrogen and thus, is an important intermediate during
the breakdown of organic matter in anoxic rice paddy soils and lake sediments. Formate is a
common substrate for methanogenesis, homoacetogenesis and sulfate reduction. However,
how much these processes contribute to formate degradation and fractionate carbon stable
isotopes is largely unknown. Therefore, we measured the conversion of formate to acetate,
$CH_4$ and $CO_2$ and the $\delta^{13}C$ of these compounds in samples of paddy soils from Vercelli
(Italy) and the International Rice Research Institute (IRRI, the Philippines) and of sediments
from the NE and SW basins of Lake Fuchskuhle (Germany). The samples were suspended in
phosphate buffer (pH 7.0) both in the absence and presence of sulfate (gypsum) and of
methyl fluoride ($CH_3F$), an inhibitor of aceticlastic methanogenesis. In the paddy soils,
formate was mainly converted to acetate both under methanogenic and sulfidogenic
conditions. Methane was only a minor product and was mainly formed from the acetate. In
the lake sediments, the product spectrum was similar, but only under methanogenic
conditions. In the presence of sulfate, however, acetate and $CH_4$ were only minor products.
The isotopic enrichment factors ($\varepsilon_{form}$) of formate consumption, determined by Mariotti plots,
were in the low range of -8‰ to -2.5‰ when sulfate was absent and formate was mainly
converted to acetate and $CH_4$. However, no enrichment factor was detectable when formate
was degraded with sulfate to mainly $CO_2$. The $\delta^{13}C$ of acetate was by about 25-50‰ more
negative than that of formate indicating acetate production by chemolithotrophic
homoacetogenesis. Hence, formate seems to be an excellent substrate for homoacetogenesis
in anoxic soils and sediments, so that this process is competing well with methanogenesis and
sulfate reduction.

**1 Introduction**
Formate is energetically almost equivalent to $H_2$ (Schink et al. 2017) and thus, is an
important intermediate in the anaerobic degradation of organic matter. Formate is a product
of microbial fermentation, where it is for example produced in pyruvate cleavage by pyruvate
formate lyase (Thauer et al., 1977) or by reduction of $CO_2$ (Schuchmann and Müller, 2013).
Formate can also be produced in secondary fermentation, such as oxidation of butyrate or
propionate (Dong et al., 1994; Sieber et al., 2014). In fact, formate and $H_2$ may equivalently
be used as electron shuttles between secondary fermenting bacteria and methanogens
(Montag and Schink, 2018; Schink et al., 2017)
Formate can serve alternatively to $H_2$ as a substrate for methanogenesis (Zinder, 1993),
(homo)acetogenesis (Drake, 1994) or sulfate reduction (Widdel, 1988), i.e.:
$4\ HCOOH \rightarrow CH_4 + 3\ CO_2 + 2\ H_2O$ (1)
$4\ HCOOH \rightarrow CH_3COOH + 2\ CO_2 + 2\ H_2O$ (2)



4 $HCOOH + SO_4^{2-} + H^+ \rightarrow HS^- + 4\ CO_2 + 4\ H_2O$       (3)
Formate may also be a substrate for syntrophic bacteria, which live from the little Gibbs free
energy ($\Delta G^{0'}$ = -3.4 kJ mol$^{-1}$) that is generated by the conversion of formate to $H_2$ plus $CO_2$
(Dolfing et al., 2008; Kim et al., 2010; Martins et al., 2015), i.e.
$HCOOH \rightarrow CO_2 + H_2$       (4)
Formate can also be enzymatically equilibrated with $H_2$ and $CO_2$ without energy generation.
This reaction happens in any organism possessing the suitable enzymes, such as formate
hydrogen lyase or hydrogen-dependent carbon dioxide reductase, and in anoxic sediments
(DeGraaf and Cappenberg, 1996; Peters et al., 1999; Schuchmann et al., 2018):
$HCOOH \leftrightarrow CO_2 + H_2$       (5)
Formate has been identified as an important substrate for methanogenesis,
homoacetogenesis or sulfate reduction in lake sediments (DeGraaf and Cappenberg, 1996;
Lovley and Klug, 1982; Phelps and Zeikus, 1985), soils (Kotsyurbenko et al., 1996; Küsel
and Drake, 1999; Rothfuss and Conrad, 1993), mires (Hausmann et al., 2016; Hunger et al.,
2011; Liebner et al., 2012; Wüst et al., 2009) and marine sediments (Glombitza et al., 2015).
However, it is not very clear to which extent formate-dependent methanogenesis,
homoacetogenesis and sulfate reduction are actually operative and to which extent formate
affects stable carbon isotope fractionation. The $\delta^{13}C$ values of compounds involved in the
degradation process of organic matter provide valuable information on the metabolic
pathways involved (Conrad, 2005; Elsner et al., 2005; Hayes, 1993). However, for correct
interpretation the knowledge of the enrichment factors ($\varepsilon$) of the major metabolic processes is
also important. The $\varepsilon$ values of methanogenesis or homoacetogenesis from $H_2$ plus $CO_2$ are
large (Blaser and Conrad, 2016). However, our knowledge of carbon isotope fractionation
with formate as substrate is scarce. In cultures of homoacetogenic bacteria the carbon in the
acetate produced from formate was strongly depleted in $^{13}C$ ($\varepsilon$ = -56.5‰) almost similarly as
with $CO_2$ as carbon source (Freude and Blaser, 2016). However, it is not known which
enrichment factors operate in methanogenic or sulfidogenic environmental samples.
Therefore, we measured isotope fractionation in methanogenic and sulfidogenic rice paddy
soils and lake sediments amended with formate. We recorded the consumption of formate
along with the production of acetate, $CH_4$ and $CO_2$ and measured the $\delta^{13}C$ of these
compounds. We also used the treatment with methyl fluoride ($CH_3F$) to inhibit the
consumption of acetate by methanogenic archaea (Janssen and Frenzel, 1997). We used the
same environmental samples as for the study of carbon isotope fractionation during
consumption of acetate (Conrad et al., 2021) and propionate (Conrad and Claus, 2023), i.e.,
rice paddy soils from Vercelli, Italy and the International Rice Research Institute (IRRI, the
Philippines) and sediments from the NE and SW basins of Lake Fuchskuhle (Germany). The
molecular data characterizing the microbial community compositions in these samples are
found in Conrad et al. ( 2021).



## 2 Materials and Methods

### 2.1 Environmental samples and incubation conditions

The soil samples were from the research stations in Vercelli, Italy and the International Rice research Institute (IRRI) in the Philippines. Sampling and soil characteristics were described before (Liu et al., 2018). The lake sediments (top 10 cm layer) were from the NE and SW basins of Lake Fuchskuhle (Casper et al., 2003). They were sampled in July 2016 using a gravity core sampler as described before (Kanaparthi et al., 2013).

The experimental setup was exactly the same as during previous studies of acetate consumption (Conrad et al., 2021) and propionate consumption (Conrad and Claus, 2023). For methanogenic conditions, paddy soil was mixed with autoclaved anoxic $H_2O$ (prepared under $N_2$) at a ratio of 1:1 and incubated under $N_2$ at 25°C for 4 weeks. In a second incubation, for sulfidogenic conditions, paddy soil was mixed with autoclaved anoxic $H_2O$ at a ratio of 1:1, was amended with 0.07 g $CaSO_4.2H_2O$, and then incubated under $N_2$ at 25°C for 4 weeks. These two preincubated soil slurries were sampled and stored at -20°C for later molecular analysis (see data in Conrad et al. ( 2021)). The preincubated soil slurries were also used (in 3 replicates) for the following incubation experiments. Two different sets of incubations were prepared. In the first set (resulting in methanogenic conditions), 5 mL soil slurry preincubated without sulfate was incubated at 25°C with 40 mL of 20 mM potassium phosphate buffer (pH 7.0) in a 150-mL bottle under an atmosphere of $N_2$. The bottles were the amended with (i) 5 mL $H_2O$; (ii) 5 mL $H_2O$ + 4.5 mL $CH_3F$; (iii) 5 mL 200 mM sodium formate; (iv) 5 mL 200 mM sodium formate + 4.5 mL $CH_3F$. In the second set (resulting in sulfidogenic conditions), 5 mL soil slurry preincubated with sulfate was incubated at 25°C with 40 mL of 20 mM potassium phosphate buffer (pH 7.0) in a 150-mL bottle under an atmosphere of $N_2$. The amendments were the same as above, but with the addition of 200 µl of a $CaSO_4$ suspension corresponding to a concentration of 2.5 M (giving a final concentration of 10 mM sulfate).

For lake sediments under methanogenic conditions, 5 ml sediment was incubated in 3 replicates at 10°C (which is close to the in-situ temperature) with 40 ml of 20 mM potassium phosphate buffer (pH 7.0) in a 150-ml bottle under an atmosphere of $N_2$. The bottles were the amended with (i) 5 ml $H_2O$; (ii) 5 ml $H_2O$ + 4.5 ml $CH_3F$; (iii) 5 ml 200 mM sodium formate; (iv) 5 ml 200 mM sodium formate + 4.5 ml $CH_3F$. For sulfidogenic conditions, lake sediments were preincubated with sulfate by adding 0.1 g $CaSO_4.2H_2O$ (gypsum) to 50 ml sediment and incubating at 10°C for 4 weeks. For sulfidogenic conditions, 5 ml of the preincubated sediment was incubated in 3 replicates at 10°C with 40 ml of 20 mM potassium phosphate buffer (pH 7.0) in a 150-ml bottle under an atmosphere of $N_2$. The bottles were amended as above, but in addition also with 200 µl of a $CaSO_4$ suspension giving a final concentration of 10 mM sulfate. Samples for later molecular analysis were taken from the



original lake sediment and from the lake sediment preincubated with sulfate. The samples
were stored at -20°C (see data in Conrad et al. ( 2021)).

*2.2 Chemical and isotopic analyses*

Gas samples for analysis of partial pressures of $CH_4$ and $CO_2$ were taken from the

headspace of the incubation bottles after vigorous manual shaking for about 30 s using a gas-
tight pressure-lock syringe, which had been flushed with $N_2$ before each sampling. Soil
slurries were sampled, centrifuged and filtered through a 0.2 μm cellulose membrane filter
and stored frozen at -20ºC for later fatty acid analysis. Chemical and isotopic analyses were
performed as described in detail previously (Goevert and Conrad, 2009). Methane was
analyzed by gas chromatography (GC) with flame ionization detector. Carbon dioxide was
analyzed after conversion to $CH_4$ with a Ni catalyst. Stable isotope analyses of $^{13}C/^{12}C$ in gas
samples were performed using GC-combustion isotope ratio mass spectrometry (GC-C-
IRMS). Formate and acetate were measured using high-performance liquid chromatography
(HPLC) linked via a Finnigan LC IsoLink to an IRMS. The isotopic values are reported in the
delta notation ($\delta^{13}C$) relative to the Vienna Peedee Belemnite standard having a $^{13}C/^{12}C$ ratio
($R_{standard}$) of 0.01118: $\delta^{13}C = 10^3 (R_{sample}/R_{standard} - 1)$. The precision of the GC-C-IRMS was
$\pm$ 0.2‰, that of the HPLC-IRMS was $\pm$ 0.3‰.

*2.3 Calculations*

Millimolar concentrations of $CH_4$ were calculated from the mixing ratios (1 ppmv = $10^{-6}$

bar) measured in the gas phase of the incubation bottles: 1000 ppmv $CH_4$ correspond to 0.09
μmol per mL of liquid. Note, that this is the total amount of $CH_4$ in the gas phase relative to
the liquid phase.

Fractionation factors for reaction A ➜ B are defined after Hayes (Hayes, 1993) as:

$\alpha_{A/B} = (\delta_A + 1000)/(\delta_B + 1000)$           (7)
also expressed as $\varepsilon \equiv 1000 (1 - \alpha)$ in permil. The carbon isotope enrichment factor $\varepsilon_{form}$
associated with formate consumption was calculated from the temporal change of $\delta^{13}C$ of
formate as described by Mariotti et al. (Mariotti et al., 1981) from the residual reactant
$\delta_r = \delta_{ri} + \varepsilon [\ln(1-f)]$           (8)
where $\delta_{ri}$ is the isotopic composition of the reactant (formate) at the beginning, and $\delta_r$ is the
isotopic composition of the residual formate, both at the instant when $f$ is determined. $f_{form}$ is
the fractional yield of the products based on the consumption of formate ($0 < f_{form} < 1$).
Linear regression of $\delta^{13}C$ of formate against $\ln(1 - f)$ yields $\varepsilon_{form}$ as the slope of best fit lines.
The regressions of $\delta^{13}C$ of formate were done for data in the range of $f_{form} < 0.7$. The linear
regressions were done individually for each experimental replicate (n = 3) and were only
accepted if $r^2 > 0.7$. The $\varepsilon$ values resulting from the replicate experiments were then averaged
($\pm$ SE).




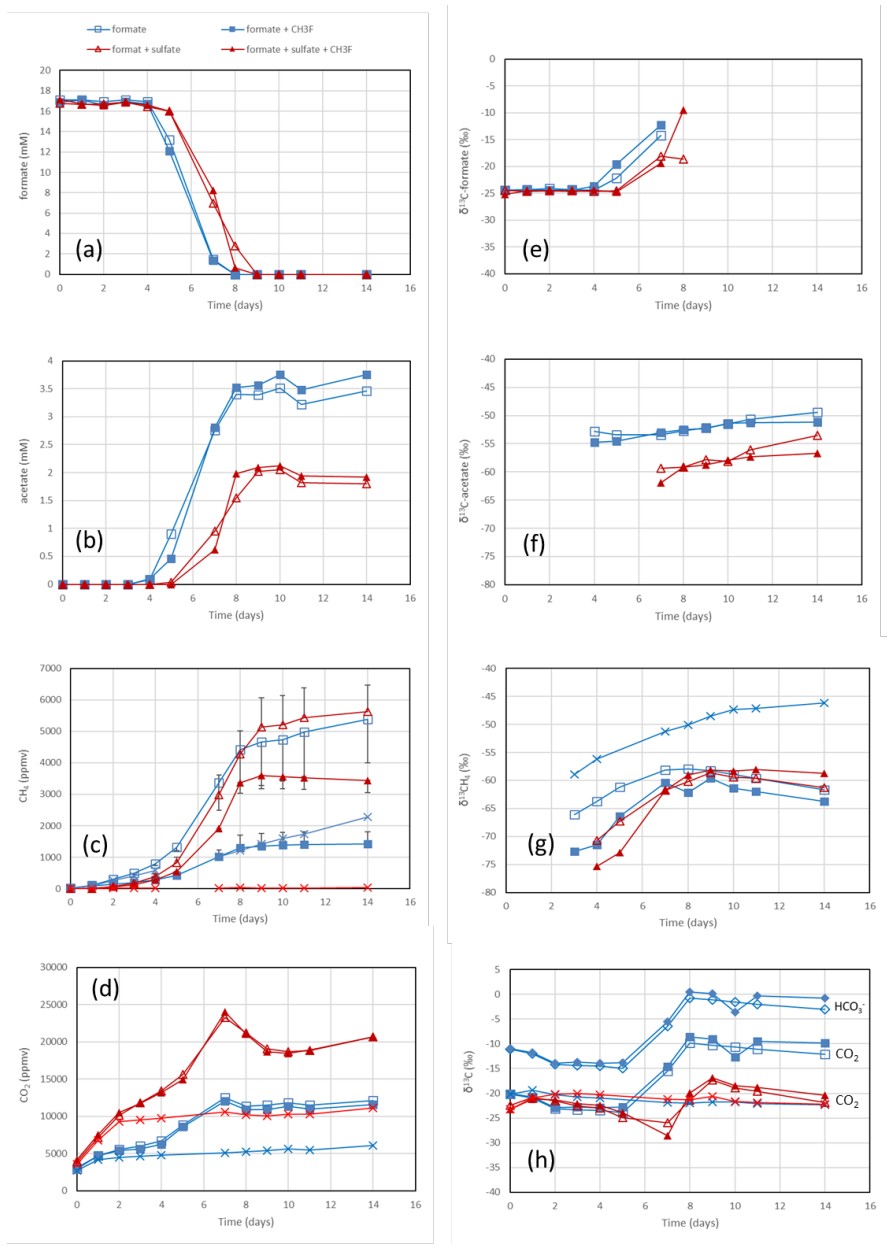


**Figure 1.** Formate conversion to acetate, $CH_4$ and $CO_2$ in suspensions of paddy soil from
Vercelli (Italy) after addition of formate without sulfate (blue squares) or formate plus sulfate
(gypsum) (red triangles) without $CH_3F$ (open symbols) or with $CH_3F$ (closed symbols).
Controls with addition of only water (blue or red X crosses) are only shown occasionally. The
panels show the temporal change of (a) concentrations of formate, (b) concentrations of
acetate, (c) mixing ratios of $CH_4$ (1 ppmv = $10^{-6}$ bar), (d) mixing ratios of $CO_2$, (e) $\delta^{13}C$ of
formate, (f) $\delta^{13}C$ of acetate, (g) $\delta^{13}C$ of $CH_4$, and (h) $\delta^{13}C$ of $CO_2$. Means ± SE.





**3 Results**
*3.1 Conversion of formate under methanogenic and sulfidogenic conditions*
The rice paddy soils were submerged and preincubated to create methanogenic or
sulfidogenic conditions. Samples of these soils were suspended in buffer at pH 7 and
amended with formate. In the Vercelli soil, formate was consumed after a lag phase of 4 days
under methanogenic and 5 days under sulfidogenic conditions (Fig. 1a). During this time the
pH increased from pH 7 up to pH 8 despite buffering. Formate consumption was not inhibited
by $CH_3F$ (Fig. 1a). Similar results were obtained with IRRI soil (Fig. S1). Acetate was
produced concomitantly with formate consumption, again without effect by $CH_3F$ (Fig. 1b).
The production of acetate under sulfidogenic conditions was smaller than under
methanogenic conditions. Methane was also produced under both methanogenic and
sulfidogenic conditions concomitantly with formate consumption (Fig. 1c; S1c). It is
noteworthy that $CH_3F$ inhibited the production of $CH_4$ (Fig. 1c; S1c). Finally, $CO_2$ was
produced under all conditions without lag phase and without effect by $CH_3F$ (Fig. 1c). In
Vercelli soil, $CO_2$ production was about twice under sulfidogenic than under methanogenic
conditions (Fig. 1c). In IRRI soil, it was only slightly larger (Fig. S1c). The accumulation of
acetate plus $CH_4$ was equimolar to the consumption of formate in terms of electron
equivalents, while the accumulation of $CH_4$ alone accounted only for <30%, in the presence
of $CH_3F$ even less (Fig. 2a; S2a). Hence, acetate was the more important product of formate
consumption. Under sulfidogenic conditions, accumulation of acetate plus $CH_4$ was less than
equimolar, especially in Vercelli soil (Fig. 2b), probably since formate was instead converted
to $CO_2$. However, acetate formation was still substantial accounting for 60-80% of formate
consumption (Fig. 2b; S2b).
The sediments from Lake Fuchskuhle were methanogenic in-situ so that preincubation of
the samples was not required. However, sulfidogenic conditions were created analogously to
the paddy soils by preincubtion with sulfate (gypsum). Substantial formate depletion did not
start before about 20 days of incubation both in sediments from the NE basin (Fig. 3) and the
SW basin (Fig. S3). Again, $CH_3F$ only inhibited the production of $CH_4$ but not that of acetate
or $CO_2$ (Fig. 3; S3). The main difference to the paddy soils was that $CH_4$ was not produced
concomitantly with formate consumption, but started right from the beginning. However, the
amounts of $CH_4$ produced were only small and were apparently due to the little formate that
was consumed in the beginning of incubation (i.e., before day 20), as seen by the fact that
$CH_4$ production in the water control (not amended with formate) was negligible (Fig. 3c;
S3c). Production of $CO_2$ started without lag phase but accelerated together with formate
consumption (Fig. 3d; S3d). In the lake sediments, $CH_4$ accounted only for <10% of formate
consumption, while acetate was the main product when sulfate was absent (Fig. 4a, S4a). In
contrast to the paddy soils, formate consumption in both lake sediments was much slower
under sulfidogenic than under methanogenic conditions (Fig. 3a; S3a). In the sediment from




SW basin, formate consumption was very slow so that less than half of the formate was
consumed during 80 days of incubation and consumption was not completed until the end of
the experiment (Fig. S3a). Very little acetate was produced and no $CH_4$ was formed from
formate in both lake sediments, when sulfate was present (Fig. 4b, S4b).

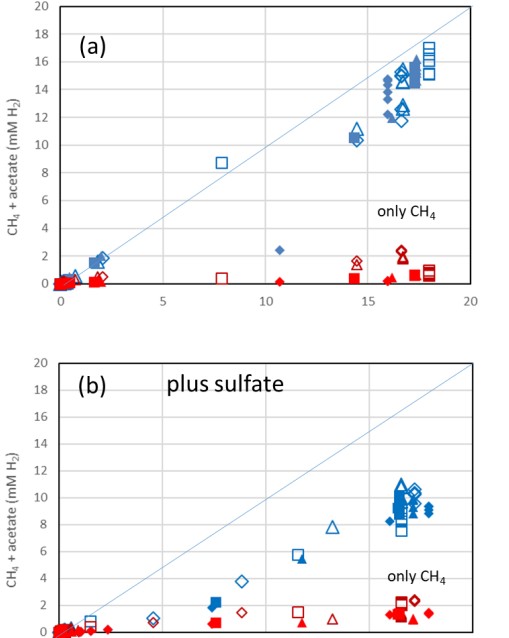

**Figure 2.** Balance of produced acetate plus $CH_4$ (blue symbols) and of only $CH_4$ (red
symbols) against the consumed formate in (a) the absence and (b) the presence of sulfate in
paddy soil from Vercelli (Italy). The open and closed symbols denote conditions in the
absence and the presence of $CH_3F$, respectively. The different symbols indicate three
different replicates. The line indicate equimolarity (in terms of reducing equivalents between
substrate and product.
*3.2 Isotope fractionation during formate consumption*
In the rice paddy soils values of $\delta^{13}C$ increased when formate was being consumed
indicating discrimination against the heavy carbon isotope. This process was not affected by
$CH_3F$ and was similar without and with sulfate (Fig. 1e; S1e). The same was the case with the
sediment from the NE lake basin, but only in the absence of sulfate (Fig. 3e). With sulfate,
the $\delta^{13}C$ of formate slowly decreased with time (Fig. 3e). In the sediment from the SW basin,
$\delta^{13}C$ of formate slowly decreased (without sulfate) or stayed constant with time (with sulfate)
(Fig. S3e). Note that formate was not completely consumed in the SW sediment when sulfate
was present (Fig. S3a).

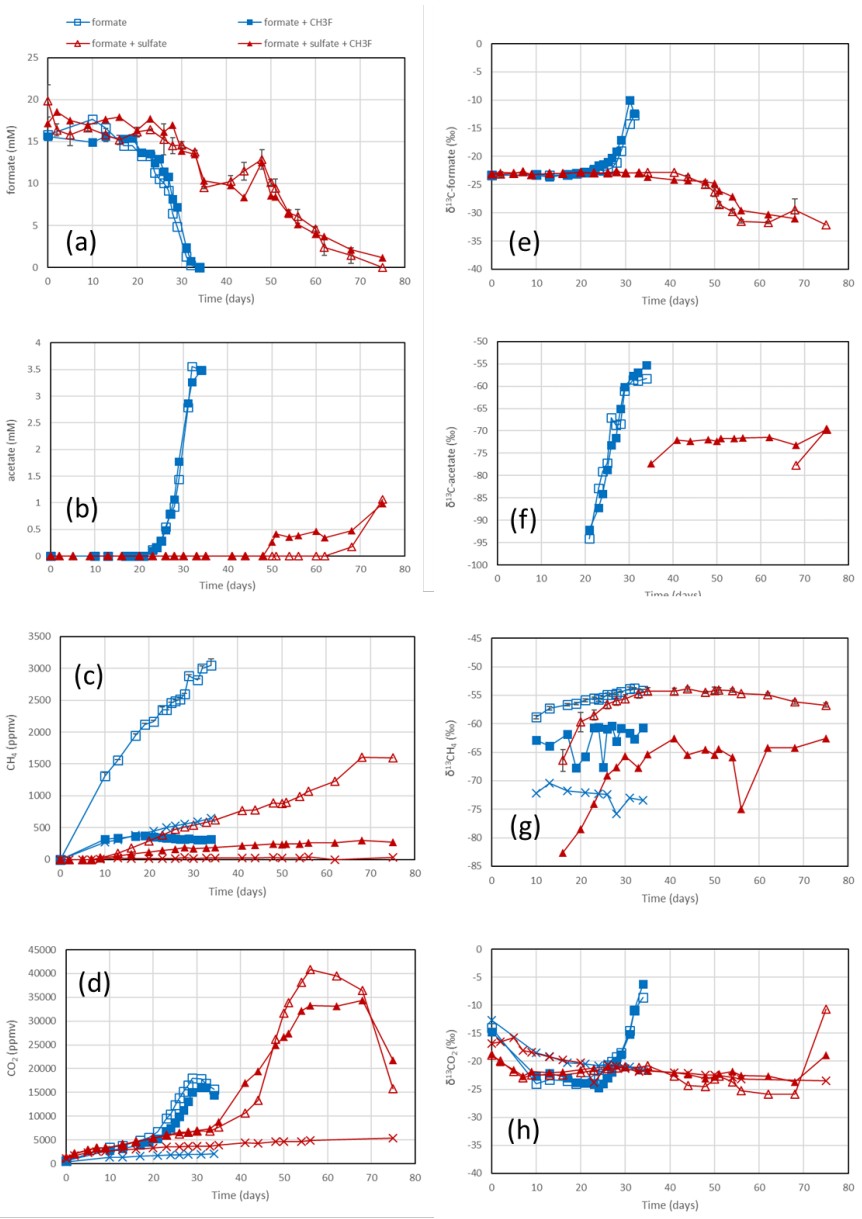

**Figure 3.** Formate conversion to acetate, $CH_4$ and $CO_2$ in suspensions of sediment from the NE basin of Lake Fuchskuhle after addition of formate without sulfate (blue squares) or formate plus sulfate (gypsum) (red triangles) without $CH_3F$ (open symbols) or with $CH_3F$ (closed symbols). Controls with addition of only water (blue or red X crosses) are only shown occasionally. The panels show the temporal change of (a) concentrations of formate, (b) concentrations of acetate, (c) mixing ratios of $CH_4$ (1 ppmv = $10^{-6}$ bar), (d) mixing ratios of $CO_2$, (e) $\delta^{13}C$ of formate, (f) $\delta^{13}C$ of acetate, (g) $\delta^{13}C$ of $CH_4$, and (h) $\delta^{13}C$ of $CO_2$. Means ± SE.

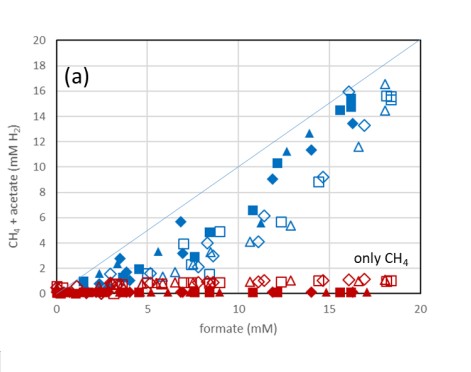

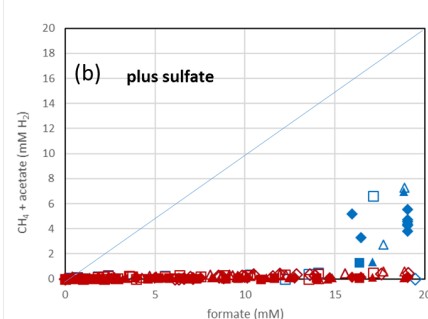

**Figure 4.** Balance of produced acetate plus $CH_4$ (blue symbols) and of only $CH_4$ (red symbols) against the consumed formate in (a) the absence and (b) the presence of sulfate in sediment from the NE basin of Lake Fuchskuhle. The open and closed symbols denote conditions in the absence and the presence of $CH_3F$, respectively. The different symbols indicate three different replicates. The line indicate equimolarity (in terms of reducing equivalents between substrate and product.

Mariotti plots of $\delta^{13}C$ of formate as function of $f_{form}$ resulted in negative slopes (Fig. 4; S5). Hence, the enrichment factors ($\varepsilon_{form}$) for the paddy soils, both without and with sulfate, and for the sediments from the NE basin of Lake Fuchskuhle without sulfate showed that the light isotope of formate carbon was preferred. Values of $\varepsilon_{form}$ were in the range of -8.5 to -2.5‰ (Fig. 6). Under sulfidogenic conditions, however, the Mariotti plots of the sediments from the NE basin (Fig. 5) did not show a negative slope and $\varepsilon_{form}$ could not be determined. The same was the case for the sediments from the SW basin (Fig. 6).

The negative $\varepsilon_{form}$ indicates that products of formate should be depleted in $^{13}C$. Indeed the $\delta^{13}C$ of acetate and $CH_4$ were generally more negative than the $\delta^{13}C$ of formate. This was the case in the paddy soils from Vercelli (Fig. 1f) and the IRRI (Fig. S1f) as well as in the sediments from the NE basin (Fig. 3f) and the SW basin (Fig. S3f) of Lake Fuchskuhle. In the sediment of the NE basin, the $\delta^{13}C$ of acetate increased from very low -95‰ to finally about -57‰ in parallel with formate consumption (Fig. 3f). $CO_2$ was also produced during formate degradation to various extent (equ.1, 2 and 3). Since the pH was in a range of pH 7 to pH 8, $CO_2$ was also converted to bicarbonate. The $\delta^{13}C$ of bicarbonate is generally by about 10‰



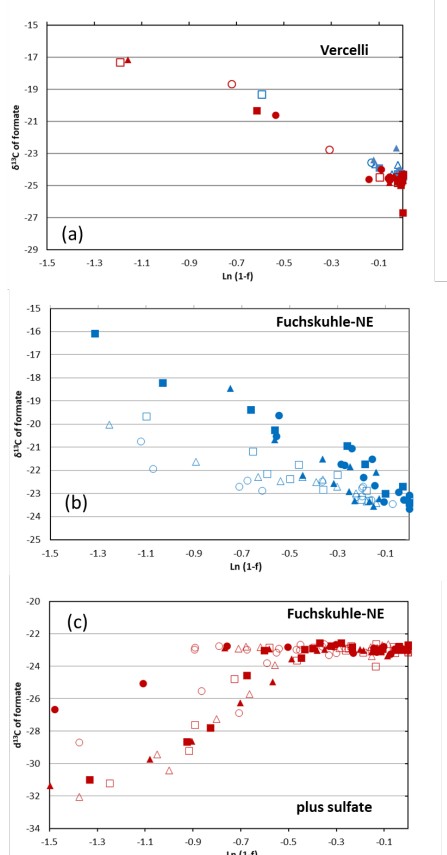

**Figure 5.** Mariotti plots of formate consumption in (a) paddy soil from Vercelli and (b, c)
sediment from the NE basin of Lake Fuchskuhle under methanogenic (blue symbols) and
sulfidogenic (red symbols) conditions both in the absence (open symbols) and in the presence
(closed symbols) of $CH_3F$. The different symbols indicate three different replicates.

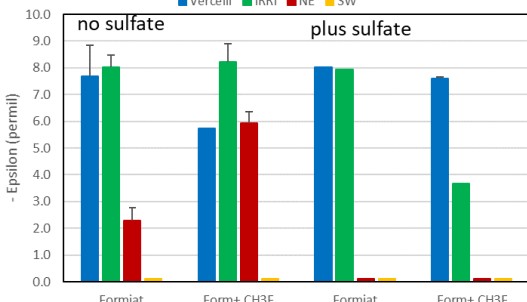

**Figure 6.** Isotopic enrichment factors ($\varepsilon_{form}$, given as negative values) in paddy soils without
and with addition of sulfate (gypsum) and $CH_3F$. Means ± SE.




more positive than the $\delta^{13}C$ of $CO_2$ (Stumm and Morgan, 1996). The $\delta^{13}C$ of the gaseous $CO_2$
was always close to the $\delta^{13}C$ of formate or was more positive. In the paddy soils and the NE
basin of Lake Fuchskuhle, the $\delta^{13}C$ of $CO_2$ increased in parallel with the increasing $\delta^{13}C$ of
formate (Fig. 1h, 3h; S1h). The $\delta^{13}C$ of the gaseous $CO_2$ produced from the formate-amended
samples was initially more negative than that from the unamended samples, but eventually
the $\delta^{13}C$ increased above these values when formate was completely consumed (Fig. 1h, 3h;
S3h).

The $\delta^{13}C$ values of the initial formate were about -24‰ (Fig. 5). When formate was

completely consumed, the $\delta^{13}C$ values of the products acetate and $CH_4$ were always more
negative. The average $\delta^{13}C$ values of the products after complete consumption of formate are
shown in Fig. 7. In the absence of sulfate, $\delta^{13}C$ of acetate was in a range of -51‰ to -49‰
and -70‰ to -63‰, in the paddy soils and lake sediments, respectively (Fig. 7). In the
presence of sulfate, $\delta^{13}C$ of acetate was in a range of -57‰ to -52‰ and -78‰ to -72‰, in
the paddy soils and lake sediments (only NE basin), respectively (Fig. 7). The $\delta^{13}C$ of $CH_4$
was in a range of -70‰ to -54‰ and -60‰ to -54‰, in the absence and presence of sulfate,
respectively (Fig. 7). The $\delta^{13}C$ of gaseous $CO_2$ (for bicarbonate plus 10‰) was in a range of -
23‰ to -11‰ and -24‰ to -19‰, in the absence and presence of sulfate, respectively (Fig.

7).


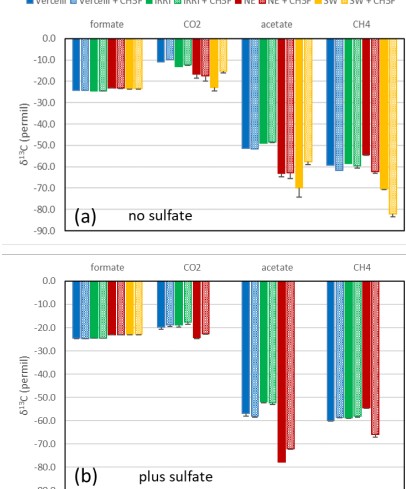


**Figure 7.** Average $\delta^{13}C$ of formate (at the beginning of incubation) and of $CO_2$, acetate and
$CH_4$ (after the depletion of formate) in soils or sediments from Vercelli (blue), the IRRI
(green), the NE basin (red) and the SW basin (yellow) in the absence (filled bars) and the
presence (dotted bars) of $CH_3F$. Means ± SE.




**4 Discussion**

*4.1 Formate degradation under acetogenic/methanogenic conditions*

In rice paddy soils formate was consumed within <10 days. The absence of sulfate did not allow sulfidogenic (equ.3) degradation, but allowed the operation of methanogenic (equ.1), homoacetogenic (equ.2) or syntrophic (equ.4) degradation. Syntrophic degradation is still disputed, since many microorganisms are able to enzymatically equilibrate $H_2$ and formate and thus prohibit generation of energy (Montag and Schink, 2018; Schink et al., 2017). Syntrophic formate degradation generates only a few kilojoules of Gibbs free energy per mole and requires the coupling with methanogenesis or other efficient hydrogen (electron) scavengers. Although formate-driven $CH_4$ production was observed in our study, the production was sensitive to inhibition by $CH_3F$ indicating that $CH_4$ was predominantly produced from acetate rather than from $H_2$. Therefore, syntrophic formate oxidation coupled to $CH_4$ production was probably not a major pathway.

Acetate was the most important product of formate degradation in the paddy soils as well as in the lake sediments. Methane also was a product, but was much less important than acetate. Furthermore, it was predominantly produced from acetate as shown by the inhibition by $CH_3F$ and the concomitant decrease of $\delta^{13}C$ of $CH_4$, which is characteristic for hydrogenotrophic methanogesis that is not inhibited by $CH_3F$ (Conrad et al., 2010). Hence, formate was apparently primarily degraded by homoacetogenesis (equ.1). Only part of the produced acetate was immediately used by aceticlastic methanogenesis generating $CH_4$ as secondary product. Although formate is a perfect substrate for homoacetogenic bacteria operating the Wood-Ljungdahl pathway (WLP) (Drake, 1994), the yield of Gibbs free energy per mole formate is less for homoacetogenic than for methanogenic degradation (Dolfing et al., 2008). Thus, it is surprising that formate-driven homoacetogenesis prevailed over methanogenesis. Nevertheless, simultaneous operation of homoacetogenesis and methanogenesis from formate has been observed before in a fen soil (Hunger et al., 2011). Homoacetogenesis prevailing over methanogenesis has also frequently been observed with $H_2/CO_2$ as substrate (Conrad et al., 1989; Nozhevnikova et al., 1994), indicating that homoacetogens can take particular advantage from low temperatures (Conrad, 2023) or the availability of secondary substrates (Peters et al., 1998).

The $\delta^{13}C$ of the produced acetate was by about 24-33‰ lower than that of formate. This isotopic discrimination between formate and acetate is similar to that measured in a culture of the homoacetogen *Thermoanaerobacter kivui* (Freude and Blaser, 2016). However, this discrimination is much larger than the isotopic enrichment factors ($\varepsilon_{form}$ of -8‰ to -2.5‰) determined from the change of $\delta^{13}C$ during formate consumption. There are two conceivable explanations for this observation. (1) Formate is disproportionated to $CO_2$ and acetate. In the WLP three formate are oxidized to $CO_2$, one formate is reduced to the methyl group of



acetate and one of the produced $CO_2$ is reduced to the carboxyl group of acetate. The
disproportionation of formate to acetate and 2 $CO_2$ is possibly a branch point (Fry, 2003;
Hayes, 2001), at which the carbon flow is split into the production of [13]C-enriched $CO_2$ and
[13]C-depleted acetate, which together result in the $\varepsilon_{form}$ observed. (2) Formate first is
completely converted to $CO_2$ plus $H_2$ (equ.5) or other electron equivalents This reaction
displays the $\varepsilon_{form}$ determined by the Mariotti plots. Acetate is then produced via the WLP by
the chemolithotrophic reduction of 2 $CO_2$ to acetate, of which the isotopic enrichment factor
is typically on the order of about -55‰ (Blaser and Conrad, 2016). In any case, it is plausible
to assume that acetate was formed via the WLP. In the WLP, oxidation of formate is
catalyzed by a formate dehydrogenase, which provides $CO_2$ to the carboxyl branch of the
WLP. The methyl branch of the WLP normally starts with formate being converted to
formyl-THF. However, it can also start with the reduction of $CO_2$ to formate with a
hydrogen-dependent carbon dioxide reductase (HDCD).  Homoacetogens (e.g., *Acetobacter*
*woodii, T. kivui*) contain such a HDCD, which allows the interconversion of formate and $H_2$
plus $CO_2$ (Jain et al., 2020; Schuchmann et al., 2018). The isotope discrimination in our
experiments indicates that the $CO_2$ produced from formate has been enriched in [13]C rather
than depleted, thus supporting the first explanation. The $\delta^{13}C$ of $CO_2$ produced from formate
was initially lower than that of the unamended soil or sediment being on the order of -20‰ to
-10‰ (Fig. 1h, 3h, S1h, S3h). Eventually, however, $\delta^{13}C$ of $CO_2$ reached values of -25‰ to -
10‰ (Fig. 7). The $\delta^{13}C$ of bicarbonate is 10‰ more positive than that of $CO_2$. This mixed
inorganic carbon would be the $CO_2$ substrate for WLP, which together with formate generates
the acetate having a $\delta^{13}C$ of about -70‰ to -50‰ (Fig. 7).
Methane was a minor product of formate degradation in all soils and sediments. Since $CH_4$
formation was strongly inhibited by $CH_3F$, it was most likely produced from acetate by
aceticlastic methanogens. Since $CH_4$ production from the soils or sediments was much lower
without formate amendment, the $CH_4$ must have primarily been produced from the acetate
that was generated from formate. The $\delta^{13}C$ of $CH_4$ in the soil incubations was more negative
than that of acetate (Fig. 7). The difference between the $\delta^{13}C$ of $CH_4$ and the $\delta^{13}C$ of acetate
indicated an isotopic enrichment factor of $\varepsilon_{ac-CH4}$ = -10‰ to -8‰, which is close to the
enrichment factor of aceticlastic *Methanosaeta (Methanothrix) concilii* (Penning et al., 2006).
In the lake sediments, the $\delta^{13}C$ of $CH_4$ and acetate were not much different indicating that
acetate was instantaneously consumed by methanogens as it was produced by homoacetogens
so that carbon isotopes were not discriminated. Both, paddy soils and lake sediments
contained *mcrA* genes (coding for a subunit of methyl CoM reductase) of *Methanosaetaceae*
*(Methanotrichaceae)* (Conrad et al., 2021).



*4.2 Formate degradation under sulfidogenic conditions*
In the rice paddy soils, formate was consumed within ten days when sulfate was present,
not quite as fast as without sulfate. In the lake sediments, however, sulfidogenic formate
consumption was much slower. Formate degradation by sulfate reduction normally results in
complete oxidation to $CO_2$ (equ.3). In the lake sediments, $CO_2$ was indeed the main
degradation product. However, in the paddy soils substantial amounts of acetate and even
$CH_4$ were also produced. The homoacetogenic bacteria in these soils apparently competed
well with the sulfate reducing bacteria, although the soils had been adapted by preincubation
in the presence of sulfate. The production of acetate and $CH_4$ was dependent on formate
degradation, since no production was observed in the unamended control. Production of $CH_4$
was inhibited by $CH_3F$ indicating that aceticlastic methanogenesis was the main process of
$CH_4$ production. The carbon isotope fractionation of formate was similar as under non-
sulfidogenic conditions, exhibiting a small $\varepsilon_{form}$ of -8‰ to -3.5‰ (Fig. 5) and displaying a
strong isotope effect with the formation of acetate ($\delta^{13}C$ = -57--52‰) and $CH_4$ ($\delta^{13}C$ = -60--
58‰). The mechanism of fractionation is probably the same (see above).
In the lake sediments, however, sulfidogenic degradation of formate was much slower
than methanogenic/acetogenic degradation. In the sediment of the SW basin, formate was not
even completely degraded within 80 days. In the sediments of both lake basins, neither
acetate nor $CH_4$ was a major product of sulfidogenic formate degradation. Hence, formate
was apparently degraded according to equ.3 forming $CO_2$ as main carbon product. This
formation process displayed no depletion of the heavy carbon isotope, as the Mariotti plots of
$\delta^{13}C$ of formate did not exhibit a negative slope. The $\delta^{13}C$ of the $CO_2$ slowly decreased with
increasing fraction of formate consumed (Fig. 3h; 5c), probably involving isotope exchange
between formate and $CO_2$ (DeGraaf and Cappenberg, 1996). The little acetate, which was
formed, displayed a $\delta^{13}C$ of -77‰ (Fig. 7b) indicating that it was produced by a similar
mechanism as in the absence of sulfate, presumably via the WLP.
The strong differences between rice paddy soils and lake sediments were possibly  caused
by their different microbial communities (Conrad et al., 2021). The differences were seen in
the composition of the *mcrA* and *dsrB* genes coding for methyl CoM reductase and
dissimilatory sulfate reductase, respectively, as well as the gene coding for the 16S rRNA.
The composition of these genes was similar whether the soils and sediments were amended
with sulfate or not. However, they were strongly different between soils and sediments
(Conrad et al., 2021). Unfortunately, these data do not allow to discriminate for particular
taxa of homoacetogenic bacteria. Nevertheless, it is possible that formate-consuming
homoacetogens were more prevalent in the soils than in the sediments and accordingly
competed more or less with the formate-consuming sulfate reducers.



*4.3 Conclusions*

Formate was found to be an excellent substrate for acetate formation in the paddy soils as
well as in the lake sediments, confirming and extending similar observations in a fen soil
(Hunger et al., 2011). In the anoxic soils, acetate was the major product even in the presence
of sulfate, which would have allowed sulfate reduction. The acetate was strongly depleted in
$^{13}C$ relative to formate, but the consumption of formate itself displayed only a small isotopic
enrichment factor. Therefore, it is likely that formate was disproportionated to $^{13}C$-depleted
acetate and $^{13}C$-enriched $CO_2$. The $\delta^{13}C$ of $CO_2$ was indeed slightly higher than that of
formate. Acetate was most likely produced by homoacetogenesis via the WLP. The produced
acetate was then used by aceticlastic methanogens (probably by *Methanothrix*), but only to
minor extent, resulting in further depletion of $^{13}C$. The homoacetogenic bacteria in the paddy
soils apparently competed well with both methanogenic and sulfate-reducing
microorganisms, when formate was the substrate. The preference of homoacetogenesis as
degradation pathway is unexpected, since other substrates, such as acetate and propionate, are
in these paddy soils degraded by methanogenesis or sulfate reduction (Conrad et al., 2021)
(Conrad and Claus, 2023). Only in the lake sediments, formate oxidation by sulfate reduction
was more prevalent than homoacetogenesis.

**Supplement link**

**Author contribution:** RC designed the experiments, evaluated the data and wrote the
manuscript. PC conducted the experiments.

**Conflicting interests:** The contact authors has declared that neither of the authors has any
competing interests.

**Financial Support**
This research has been supported by the Fonds der Chemischen Industrie (grant no. 163468).




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



**Figure legends**

**Figure 1.** Formate conversion to acetate, $CH_4$ and $CO_2$ in suspensions of paddy soil from Vercelli (Italy) after addition of formate without sulfate (blue squares) or formate plus sulfate (gypsum) (red triangles) without $CH_3F$ (open symbols) or with $CH_3F$ (closed symbols). Controls with addition of only water (blue or red X crosses) are only shown occasionally. The panels show the temporal change of (a) concentrations of formate, (b) concentrations of acetate, (c) mixing ratios of $CH_4$ (1 ppmv = $10^{-6}$ bar), (d) mixing ratios of $CO_2$, (e) $\delta^{13}C$ of formate, (f) $\delta^{13}C$ of acetate, (g) $\delta^{13}C$ of $CH_4$, and (h) $\delta^{13}C$ of $CO_2$. Means ± SE.

**Figure 2.** Balance of produced acetate plus $CH_4$ (blue symbols) and of only $CH_4$ (red symbols) against the consumed formate in (a) the absence and (b) the presence of sulfate in paddy soil from Vercelli (Italy). The open and closed symbols denote conditions in the absence and the presence of $CH_3F$, respectively. The different symbols indicate three different replicates. The line indicate equimolarity (in terms of reducing equivalents between substrate and product.

**Figure 3.** Formate conversion to acetate, $CH_4$ and $CO_2$ in suspensions of sediment from the NE basin of Lake Fuchskuhle after addition of formate without sulfate (blue squares) or formate plus sulfate (gypsum) (red triangles) without $CH_3F$ (open symbols) or with $CH_3F$ (closed symbols). Controls with addition of only water (blue or red X crosses) are only shown occasionally. The panels show the temporal change of (a) concentrations of formate, (b) concentrations of acetate, (c) mixing ratios of $CH_4$ (1 ppmv = $10^{-6}$ bar), (d) mixing ratios of $CO_2$, (e) $\delta^{13}C$ of formate, (f) $\delta^{13}C$ of acetate, (g) $\delta^{13}C$ of $CH_4$, and (h) $\delta^{13}C$ of $CO_2$. Means ± SE.

**Figure 4.** Balance of produced acetate plus $CH_4$ (blue symbols) and of only $CH_4$ (red symbols) against the consumed formate in (a) the absence and (b) the presence of sulfate in sediment from the NE basin of Lake Fuchskuhle. The open and closed symbols denote conditions in the absence and the presence of $CH_3F$, respectively. The different symbols indicate three different replicates. The line indicate equimolarity (in terms of reducing equivalents between substrate and product.

**Figure 5.** Mariotti plots of formate consumption in (a) paddy soil from Vercelli and (b, c) sediment from the NE basin of Lake Fuchskuhle under methanogenic (blue symbols) and sulfidogenic (red symbols) conditions both in the absence (open symbols) and in the presence (closed symbols) of $CH_3F$. The different symbols indicate three different replicates.

**Figure 6.** Isotopic enrichment factors ($\varepsilon_{form}$, given as negative values) in paddy soils without and with addition of sulfate (gypsum) and $CH_3F$. Means ± SE.



**Figure 7.** Average $\delta^{13}C$ of formate (at the beginning of incubation) and of $CO_2$, acetate and $CH_4$ (after the depletion of formate) in soils or sediments from Vercelli (blue), the IRRI (green), the NE basin (red) and the SW basin (yellow) in the absence (filled bars) and the presence (dotted bars) of $CH_3F$. Means ± SE.

## Legends of the supplemental figures

Fig. S1: Formate conversion to acetate, $CH_4$ and $CO_2$ in suspensions of paddy soil from the International Rice Research Institute (IRRI) after addition of formate without sulfate (blue squares) or formate plus sulfate (gypsum) (red triangles) without $CH_3F$ (open symbols) or with $CH_3F$ (closed symbols). Controls with addition of only water (blue or red X crosses) are only shown occasionally. The panels show the temporal change of (a) concentrations of formate, (b) concentrations of acetate, (c) mixing ratios of $CH_4$ (1 ppmv = $10^{-6}$ bar), (d) mixing ratios of $CO_2$, (e) $\delta^{13}C$ of formate, (f) $\delta^{13}C$ of acetate, (g) $\delta^{13}C$ of $CH_4$, and (h) $\delta^{13}C$ of $CO_2$. Means ± SE.

Fig. S2: Balance of produced acetate plus $CH_4$ (blue symbols) and of only $CH_4$ (red symbols) against the consumed formate in (a) the absence and (b) the presence of sulfate in paddy soil from the IRRI. The open and closed symbols denote conditions in the absence and the presence of $CH_3F$, respectively. The different symbols indicate three different replicates. The line indicate equimolarity (in terms of reducing equivalents between substrate and product.

Fig. S3: Formate conversion to acetate, $CH_4$ and $CO_2$ in suspensions of sediment from the SW basin of Lake Fuchskuhle after addition of formate without sulfate (blue squares) or formate plus sulfate (gypsum) (red triangles) without $CH_3F$ (open symbols) or with $CH_3F$ (closed symbols). Controls with addition of only water (blue or red X crosses) are only shown occasionally. The panels show the temporal change of (a) concentrations of formate, (b) concentrations of acetate, (c) mixing ratios of $CH_4$ (1 ppmv = $10^{-6}$ bar), (d) mixing ratios of $CO_2$, (e) $\delta^{13}C$ of formate, (f) $\delta^{13}C$ of acetate, (g) $\delta^{13}C$ of $CH_4$, and (h) $\delta^{13}C$ of $CO_2$. Means ± SE.

Fig. S4: Balance of produced acetate plus $CH_4$ (blue symbols) and of only $CH_4$ (red symbols) against the consumed formate in (a) the absence and (b) the presence of sulfate in sediment from the SW basin of Lake Fuchskuhle. The open and closed symbols denote conditions in the absence and the presence of $CH_3F$, respectively. The different symbols indicate three different replicates. The line indicate equimolarity (in terms of reducing equivalents between substrate and product.



Fig. S5: Mariotti plots of formate consumption in (a) paddy soil from the IRRI and (b, c)

sediment from the SW basin of Lake Fuchskuhle under methanogenic (blue symbols)

and sulfidogenic (red symbols) conditions both in the absence (open symbols) and in the

presence (closed symbols) of $CH_3F$. The different symbols indicate three different

replicates.