# Peer review of "Fractionation of stable carbon isotopes during formate consumption in"

_EGUsphere, 2023_

## Author Response (AR1)

**Reply to the reviewers´comments**

The following response basically is the same as that posted as final author comments. However, the yellow text describes what has been done with the manuscript and the figures.

*Reviewer #1*

In this manuscript, the authors report on carbon isotope fractionation during formate transformation in rice paddy soils and lake sediment. They observe formate utilization mainly through homoacetogenesis to be the dominant process that outcompetes methanogenesis and sulfate reduction.

As it appears, the data were collected and evaluated correctly and there is no doubt on their reliability. Nonetheless, it does not become clear why this study was performed and what the outcome is in some few clear sentences. Why was sediment from Fuchskuhle used? Because there were still some 6 years old sediment samples in the coldroom? What are the properties of this specific (and internationally not so well-known) lake (pH, carbonate and sulfate content, trophic status etc.)? Is it representative of freshwater lakes in general? Which general conclusions can be drawn from this study?

*Reply:*

The incentive for the study is stated in L74-79: "The $\varepsilon$ values of methanogenesis or homoacetogenesis from H2 plus CO2 are large (Blaser and Conrad, 2016). However, our knowledge of carbon isotope fractionation with formate as substrate is scarce. In cultures of homoacetogenic bacteria the carbon in the acetate produced from formate was strongly depleted in 13C ($\varepsilon = -56.5\text{‰}$) almost similarly as with CO2 as carbon source (Freude and Blaser, 2016). However, it is not known which enrichment factors operate in methanogenic or sulfidogenic environmental samples." For this purpose, we arbitrarily chose two different methanogenic environments, paddy soil and Lake Fuchskuhle sediments. The reason for chosing them was simply because we had used them before and had background information (e.g. gene sequences). Of course we could have chosen different environments (sewage digestor, peat), but a priori it did not matter in which samples to test for fractionation factors of formate degradation.

Lake Fuchskuhle is a lake in Northern Germany, which had been the subject of several studies, also own ones. We have expanded the description. For our purpose it was as good as any other lake. By the way, the lake sediment was sampled in 2016, and the experiments were performed in 2017. The experiments on paddy soils were done in 2016. We now have added this information to the methods section.

General conclusions: It was during the study of fractionation factors that we realized that formate degradation was dominated by homoacetogenesis, which we found to be intriguing. I must admit that we initially expected that formate would be mainly consumed by methanogenesis. However, this was not the case. We briefly discussed this observation but did not present a theoretical explanation. Now, we have added a few sentences mentioning the different molecular mechanisms of formate utilization in acetogens versus methanogens, such as outlined in the

recent review by Lemaire et al. (Frontiers Microbiol. 2020). General conclusions are that homoacetogenesis seems to be a major pathway of formate utilization in nature, and that strong fractionation of carbon isotopes occurs mainly during the formation of acetate via the Wood-Ljungdahl pathway, while the initital consumption of formate displayed only a small enrichment factor.

Minor comments:

1. 111, l. 120: Omit "the"; done.
2. 190: Sentence unclear; sentence rephrased.

Fig. 6, correct Formiat to Formate; done.

1. 307: Energy cannot be generated; sentence rephrased.
2. 342: equivalents; corrected.
3. 403: Where are the data on the 16SrRNA genes? Now mentioned.
4. 404: "composition of these genes"? What do the authors mean with that? Sentence rephrased.
5. 425: …are degraded in these paddy soils…; done.
6. 434: author; corrected.

*Reviewer #2*

*General Comments*

The manuscript by Conrad and Claus, entitled "Fractionation of stable carbon isotopes during formate consumption in anoxic rice paddy soils and lake sediments" presents an interesting study that analyzes isotope effects associated with formate consumption during incubation experiments. The experiments were conducted with two rice pady soils and two lake sediments, each in triplicate at an incubation temperature of 25°C and over a period of four weeks. The authors have used a straightforward experimental approach and state-of-the art analytical methods, including stable isotope analyses of methane by GC-IRMS and stable isotope analyses of formate and acetate by HPLC-IRMS. They observe formation of acetate from formate with a strong [13]C-depletion of acetate relative to formate, pointing to homoacetogenesis via the Wood-Ljungdahl pathway.

The scientific questions are relevant and within the scope of BG. The isotopic data are novel. They are discussed in an appropriate and balanced way, and the conclusion is substantial. I couldn't detect major scientific flaws.

My major scientific concern is the amendment of fairly large amounts of formate that produce starting concentrations of more than 15 mM in the incubation experiments. How do these concentrations compare to natural formate levels in paddy soils, lake sediments or other environments? And would formate still be an excellent substrate for acetate production via homoacetogenesis at natural concentration levels? How valid are the findings at 25°C for lake

sediments that are generally much colder under in situ conditions? These questions warrant some further discussion.

The manuscript is well organized overall, but the quality of the figures needs improvement. Clear and concise figure captions and legends could improve the readability of the manuscript a lot.

*Reply*

The formate concentrations in-situ are on the order of micromolar concentrations (Montag & Schink 2018). It is correct that relatively large amounts of formate had to be added to observe a degradation of formate combined with isotope fractionation. In-situ concentrations typically (albeit not always) are steady state concentrations between production and consumption. Under such conditions a compound is consumed after its production without exhibiting kinetic isotope fractionation, simply since everything is consumed without option to preferentially use the light isotope. In order to observe isotope fractionation, it is therefore necessary to increase the concentration of the compound to a level, where the reaction process can exhibit a preference for the light isotope. The criterion for our experiments was the fit of the isotope values of formate to the Mariotti equation, i.e., the of formate versus the logarithm of the fraction of formate utilized. This criterion was fulfilled. We assume that the observation of formate isotope fractionation according to the Wood Ljungdahl pathway would not only be valid that the elevated concentrations where isotope fractionation occurred, but also under steady state conditions, when all the newly produced formate is completely consumed. ==We have now added a short paragraph with explanation to our manuscript.==

As stated in the Experimental Section we incubated the paddy soils at 25°C, but the lake sediments at 10°C, which is the in-situ value. ==This statement has probably been overlooked by the reviewer. We have made no changes.==

*Specific comments and suggestions for the improvement of figures:*

**Figure 1:** The figure legend is incomplete. The crosses representing the controls are not present. Panels c, d and h display a varying shade of red with no clarification. In my opinion, the size of the axis title font is quite small. While the title of the x-axis is capitalized, that of the y-axis is not. In panel e, f, g, and h, it would be appropriate to add "vs VPDB" to the per mil sign on the y-axis. In panel c and d, it would be better to show molar concentrations of methane and carbon dioxide in the liquid phase rather than partial pressures in the gas phase, in particular since the conversion is explained in the methods section (L. 149-152). Partial pressures cannot be compared to the molar concentrations of acetate and formate.

==The figure legends have been amended including the crosses and the shades of red have been corrected. The font size has been increased. All titles have now been capitalized. However, we did not add vs.VPDB, since it is an acronym and since the calibration is described in the Methods. We also did not change the gas partial pressures into molar concentrations. Although this would be easy for methane (as described in the Methods), it would be difficult for carbon==

dioxide, since much of it is in dissolved state and (depending on pH) as bicarbonate. Since the molar concentrations of methane are explicitly shown in Fig. 2, and those of $CO_2$ are not relevant for the discussion, we decided to keep partial pressures (which where the actually analyzed entities).

**Figure 2:** The figure is lacking a legend. The title of the x-axis is not capitalized. Check for consistency. "The different symbols indicate three different replicates" – Well, I cannot decipher the information, and I cannot distinguish the different shades of red. The title of the y-axis gives $CH_4$ and acetate in mM H2. This needs explanation. The figure would benefit from a more concise figure caption.

A legend has now been added. The titles have now been capitalized. The equivalence of $H_2$ with $CH_4$, acetate and formate has now been explained.

**Figure 3**: The figure legend is incomplete. The crosses representing the controls are not present. For methane and carbon dioxide, it would be better to show molar concentrations in the liquid phase rather than partial pressures in the gas phase (see above).

Corrected analogously to Fig. 1

**Figure 4:** See comments on Figure 2.

Corrected analogously to Fig. 2

**Figure 5:** The figure is lacking a legend. It is confusing that results for Fuchskuhle are shown in two panels to distinguigh sulfidogenic and methanogenic conditions, while both data sets are combined in one panel for Vercelli. On the y-axis, "per mil vs VPDB" is missing.

We have now presented 2 panels for Vercelli soil and two panels for Fuchskuhle sediment. Legends have been added.

**Figure 6:** The figure legend is unsatisfactory, because it uses abbreviations that are not self-explanatory. The figure caption is not complete, because it is lacking information on the lake sediments. In my opinion, it would be better to present the information in two panels (e.g. (a) without sulfate, (b) with sulfate). Check the spelling of formate.

We have separated the Figure into two sections. The acronyms are explained in the figure caption. Spelling corrected.

**Figure 7:** The figure legend and caption need clarification as suggested for Fig. 1-6. Please add full information on sampling sites, such as rice paddy soil and lake sediment. The abbreviation "IRRI" in introduced in the text, but it would increase readability if the full name was given in the figure caption.

The figure caption has been amended accordingly.

*Other minor comments:*

L. 111: "The bottles were the amended" seems to contain a typo. Change to "The bottles were amended"; done.

L. 226: "In the rice paddy soils values of $\delta^{13}C$ increased when formate was being consumed indicating discrimination against the heavy carbon isotope." This sentence is not clear. Which $\delta^{13}C$-values increased? Change to "In the rice paddy soils, $\delta^{13}C$-values of formate increased when formate was consumed indicating discrimination against the heavy carbon isotope."; done.